# Predicting Cellular Responses to Novel Drug Perturbations at a Single-Cell Resolution

**Leon Hetzel**[*1, 3], **Simon Böhm**[*3], **Niki Kilbertus**[2, 4],
**Stephan Günnemann**[2], **Mohammad Lotfollahi**[1, 5], and **Fabian Theis**[1, 3]

{leon.hetzel, simon.boehm, niki.kilbertus}@helmholtz-muenchen.de
s.guennemann@tum.de, {mohammad.lotfollahi, fabian.theis}@helmholtz-muenchen.de

[1]Department of Mathematics, Technical University of Munich
[2]Department of Computer Science, Technical University of Munich
[3]Helmholtz Center for Computational Health, Munich
[4]Helmholtz AI, Munich
[5] Wellcome Sanger Institute, Cambridge

## Abstract

Single-cell transcriptomics enabled the study of cellular heterogeneity in response to perturbations at the resolution of individual cells. However, scaling high-throughput screens (HTSs) to measure cellular responses for many drugs remains a challenge due to technical limitations and, more importantly, the cost of such multiplexed experiments. Thus, transferring information from routinely performed bulk RNA HTS is required to enrich single-cell data meaningfully. We introduce chemCPA, a new encoder-decoder architecture to study the perturbational effects of unseen drugs. We combine the model with an architecture surgery for transfer learning and demonstrate how training on existing bulk RNA HTS datasets can improve generalisation performance. Better generalisation reduces the need for extensive and costly screens at single-cell resolution. We envision that our proposed method will facilitate more efficient experiment designs through its ability to generate in-silico hypotheses, ultimately accelerating drug discovery.

## 1 Introduction

Recent advances in single-cell methods allowed the simultaneous analysis of millions of cells, increasing depth and resolution to explore cellular heterogeneity (Sikkema et al., 2022; Han et al., 2020). With single-cell RNA sequencing (scRNA-seq) and high-throughput screens (HTSs) one can now study the impact of different perturbations, i.e., drug-dosage combinations, on the transcriptome at cellular resolution (Yofe et al., 2020; Norman et al., 2019). Unlike conventional HTSs, scRNA-seq HTSs can identify subtle changes in gene expression and cellular heterogeneity, constituting a cornerstone for pharmaceutics and drug discovery (Srivatsan et al., 2020). Nevertheless, these newly introduced sample multiplexing techniques (McGinnis et al., 2019; Stoeckius et al., 2018; Gehring et al., 2018) require expensive library preparation and do not scale to screen thousands of distinct molecules. Even in its most cost-effective version, nuclear hashing, the acquired datasets contain no more than 200 different drugs (Srivatsan et al., 2020).

Consequently, computational methods are required to address the limited exploration power of existing experimental methods and discover promising therapeutic drug candidates. Suitable methods

---

[*]equal contribution
Code is available at github.com/theislab/chemCPA.

36th Conference on Neural Information Processing Systems (NeurIPS 2022).

should predict the response to unobserved (combinations of) perturbations. Increasing in difficulty, such tasks may include inter- and extrapolation of dosage values, the generalisation to unobserved (combinations of) drug-covariates (e.g., cell-type), or predictions for unseen drugs. In terms of medical impact, the prediction of unobserved perturbations may be the most desirable, for example for drug repurposing. At the same time, it requires the model to properly capture complex chemical interactions within multiple distinct cellular contexts. Such generalisation capabilities can not yet be learned from single-cell HTSs alone, as they supposedly do not cover the required breadth of chemical interactions. In this work, we leverage information across datasets to alleviate this issue.

We propose a new model that generalises previous work on Fader Networks by Lample et al. (2017) and the Compositional Perturbation Autoencoder (CPA) by Lotfollahi et al. (2021) to the challenging scenario of generating counterfactual predictions for unseen compounds. Our method is as flexible and interpretable as CPA but further enables us to leverage lower resolution but higher throughput assays, such as bulk RNA HTSs, to improve the model's generalisation performance on single-cell data (Amodio et al., 2021). Our main contributions are:

1. We introduce chemCPA, a model that incorporates knowledge about the compounds' structure, enabling the prediction of drug perturbations at a single-cell level from molecular representations.

2. We propose and evaluate a transfer learning scheme to leverage HTS bulk RNA-seq data in the setting of both identical and different gene sets between the source (bulk) and target (single-cell) datasets.

3. We show how chemCPA outperforms existing methods on the task of predicting unobserved drug-covariate combinations. At the same time, we demonstrate chemCPA's versatility and evaluate chemCPA on generalisation tasks that cannot be modeled using any previously existing method.

## 2 Related Work

Over the past years, deep learning (DL) has become an essential tool for the analysis and interpretation of scRNA-seq data (Angerer et al., 2017; Rybakov et al., 2020; Lopez et al., 2020; Hetzel et al., 2021). Representation learning in particular, has been useful not only for identifying cellular heterogeneity and integration (Gayoso et al., 2022), or mapping query to reference datasets (Lotfollahi et al., 2022), but also in the context of modelling single-cell perturbation responses (Rampášek et al., 2019; Seninge et al., 2021; Lotfollahi et al., 2019; Ji et al., 2021).

Unlike linear models (Dixit et al., 2016; Kamimoto et al., 2020) or mechanistic approaches (Fröhlich et al., 2018; Yuan et al., 2021), DL is suited to capture non-linear cell-type-specific responses and easily scales to genome-wide measurements. Recently, Lotfollahi et al. (2021) introduced the CPA method for modelling perturbations on scRNA-seq data. CPA does not generalise to unseen compounds, hindering its application to virtual screening of drugs not yet measured via scRNA-seq data, which is required for effective drug discovery.

For bulk RNA data, on the other hand, several methods have been proposed to predict gene expression profiles for de novo chemicals (Pham et al., 2021; Zhu et al., 2021; Umarov et al., 2021). Crucially, the L1000 dataset, introduced by the LINCS programme (Subramanian et al., 2017), greatly facilitated such advances on phenotype-based compound screening. However, it remains unclear how to translate these approaches to single-cell datasets that include significantly fewer compounds and, in many cases, rely on different gene sets.

## 3 Chemical Compositional Perturbation Autoencoder

We consider a dataset $\mathcal{D} = \{(x_i, y_i)\}_{i=1}^N = \{(x_i, (d_i, s_i, c_i))\}_{i=1}^N$, where $x_i \in \mathbb{R}^n$ describes the $n$-dimensional gene expression and $y_i$ an attribute set. For scRNA-seq perturbation data, we usually consider the drug and dosage attributes, $d_i \in \{\text{drugs in } \mathcal{D}\}$ and $s_i \in \mathbb{R}$, respectively, and the cell-line $c_i$ of cell $i$. Note that this set of attributes $\mathcal{Y}$ depends on the available data and could be extended to covariates such as patient, or species.

A possible approach to predicting counterfactual combinations is to encode a cell's gene expression $x_i$ invariantly from its attributes $y_i$ as a latent vector $z_i$, called the basal state. Being provided such

(1) Encoder-Decoder:  (2) Attribute embeddings:  (3) Adversarial classifiers:

Figure 1: Architecture of chemCPA. The model consists of three parts: (1) the encoder-decoder architecure, (2) the attribute embeddings, and (3) the adversarial classifiers. The molecule encoder $G$ can be any graph- or language-based model as long as it generates fixed-sized embeddings $h_{\text{drugs}}$. The MLPs $S$ and $M$ are trained to map the embeddings to the perturbational latent space. There, $z_{d_i}$ is added to the basal state $z_i$ and the covariate embedding $z_{c_i}$. In this work, the latter always corresponds to cell lines. The basal state $z_i = E_\theta(x_i)$ is trained to be invariant through adversarial classifiers $A_\phi^j$ and the decoder $D_\psi$ gives rise to the Gaussian likelihood $\mathcal{N}(x_i \mid \mu_i, \sigma_i)$.

disentangled representations, $z_i$ can be combined with attributes, $z_{d_i}$ and $z_{c_i}$, to encode any attribute combination $y_i' \neq y_i$, and decoded back to a gene expression state $\hat{x}_i$ that corresponds to this new set of chosen attributes.

To this end, we divide our model, the Chemical Compositional Perturbation Autoencoder (chemCPA), into three parts: (1) the gene expression encoder and decoder, (2) the attribute embedders, and (3) the adversarial classifiers, see Figure 1 for an illustration.

## 3.1 Gene encoder and decoder

Following Lotfollahi et al. (2021), our model is based on an encoder-decoder architecture combined with adversarial training. The encoding network $E_\theta : \mathbb{R}^n \to \mathbb{R}^l$ is a multi-layer perceptron (MLP) with parameters $\theta$ that maps a measured gene expression state $x_i \in \mathbb{R}^n$ to its $l$-dimensional latent vector $z_i = E_\theta(x_i)$. Through adversarial classifiers, $z_i$ is trained to not contain any information about its attributes $y_i$. This gives us control over the latent space in which we update $z_i$ with an additive attribute embedding of our choice and obtain $z_i'$.

The decoder $D_\psi : \mathbb{R}^l \to \mathbb{R}^{2n}$ is an MLP that takes $z_i'$ as input and computes the component-wise parameters of the underlying distribution $\mathbb{P}$ of the gene expression data. Dependent on whether $x_i$ is raw or pre-processed, $\mathbb{P}$ can follow a negative binomial or Gaussian distribution. Assuming a mean and variance parametrisation, we get in both cases $\mu = D_\psi^\mu(z')$ and $\sigma^2 = D_\psi^{\sigma^2}(z')$ for the description of the decoded gene expression state. While chemCPA supports both settings, we observed better convergence with a Gaussian likelihood for which the reconstruction loss becomes:

$$\mathcal{L}_{\text{rec}}(\theta, \psi) = N(x_i \mid \mu_i, \sigma_i) = \frac{1}{2}\left[\ln\left(D_\psi^{\sigma^2}(z_i')\right) + \frac{\left(D_\psi^\mu(z_i') - x_i\right)^2}{D_\psi^{\sigma^2}(z_i')}\right] \text{ with } z' = E_\theta(x) + z_{\text{attribute}} \ ,$$

Next, we provide intuition about how we can meaningfully interpret latent space arithmetics and how we encode drug and cell-line attributes.

## 3.2 Attribute embedding and additive latent space

We assume an additive structure of the perturbation response in the latent space:

$$z_i' = z_i + z_{\text{attribute}} = z_i + z_{c_i} + \hat{s}_i z_{d_i} \ ,$$

where $z_{c_i}$ and $z_{d_i}$ correspond to the latent cell-line and drug attributes, and $\hat{s}_i$ encodes the dosage.

This choice of linearity makes the model interpretable for users such as biologists since it permits to analyse and ablate components individually, e.g., allowing interpolation or extrapolation of the dose values. Another advantage of the additive structure is its permutation invariance and that it allows for adding new covariates, e.g., during fine-tuning. While remaining interpretable, chemCPA is able to model complex relationships through the non-linear decoder.

Due to their different nature, we encode the drug and cell-line attributes separately in the latent space. For the cell-lines, we use the same approach as Lotfollahi et al. (2021), where a $l$-dimensional latent representation $z_c$ is optimised for each cell-line $c$. For the drugs, we propose a new embedding network $P_\varphi$.

**Perturbation Network**  The network $P_\varphi$ maps molecular representations—such as its graph or SMILES representation—and the used dosage to its latent perturbation state. This perturbation network $P_\varphi$ consists of the molecule and perturbation encoders, $G$ and $M$, as well as the dosage scaler $S$, see Figure 1 (2).

The molecule encoder $G : \mathcal{G} \to \mathbb{R}^m$ encodes the molecule representation $g_i \in \mathcal{G}$ as a fixed size embedding $h_{d_i} \in \mathbb{R}^m$. In a subsequent step, the perturbation encoder $M : \mathbb{R}^m \to \mathbb{R}^l$ takes the molecular embedding $h_{d_i}$ as input and generates the drug perturbation $z_{d_i} \in \mathbb{R}^l$ that is used in chemCPA's latent space.

The dosage scaler $S : \mathbb{R}^{m+1} \to \mathbb{R}$ also uses $h_{d_i}$ and maps it together with the dosage $s_i$ to the scaled dosage value $\hat{s}_i$. We chose $S$ to map back to a scalar value $\hat{s}$ as this allows us to compute drug-response curves in an easy fashion. In addition, this way of encoding matches the idea that $z_{d_i}$ encodes the drug's general effect, which is dosage independent. Put together, we end up with

$$\hat{s}_i \times z_{d_i} = P_\varphi(g_i, s_i) = S(h_{d_i}, s_i) \times M(h_{d_i}) \text{ with } h_{d_i} = G(g_i) \quad .$$

The molecule encoder $G$ can be any encoding network that maps molecular representations to fixed-size embeddings. Due to the limited number of drugs available in scRNA-seq HTSs, we propose to rely on a pretrained encoding model and freeze $G$ during training. We tested multiple different options for $G$ and include a detailed benchmark in the Appendix A.1. We found that RDKit features performed well in our setting and report all following results for chemCPA with RDKit as the molecule encoder $G$. By design, we can choose a new set of attributes for the drug and cell line and compute the new latent state as $z_i' = z_i + z_{\text{attribute}}$ at test time. Due to the perturbation network, chemCPA makes it possible to predict drug perturbations for molecules that have not been experimentally observed ($d \notin \mathcal{D}$). In contrast, CPA can only make predictions for molecules that were present during training ($d \in \mathcal{D}$). In both cases, the latent representation is computed as:

$$z_i' = z_i + z_{\text{attribute}} = z_i + z_{c_i} + \hat{s}_i z_{d_i} \, .$$

We next describe how we "strip $z_i$ from its attribute information" to obtain a basal state representation.

## 3.3 Adversarial classifiers for invariant basal states

To generate invariant basal states and produce disentangled representations $z_i$, $z_{d_i}$, and $z_{c_i}$, we use adversarial classifiers $A_\phi^{\text{drug}}$ and $A_\phi^{\text{cov}}$. Both adversary networks $A_\phi^j : \mathbb{R}^l \to \mathbb{R}^{N_j}$ take the latent basal state $z_i$ as input and aim to predict the drug that has been applied to example $i$ as well as its cell-line $c_i$. While these classifiers are trained to improve classification performance, we also add the classification loss *with a reversed sign* to the training objective for the encoder $E_\theta$. Hence, the encoder attempts to produce a latent representation $z_i$ *which contains no information about the attributes*. Note that this explicit separation of basal, drug, and covariate information, which we call disentanglement, is an approximation to make the problem tractable. At the same time, such separation is useful for attributing perturbation effects to specific sources, e.g., drug or cell line, which is relevant for biological applications and downstream analyses.

We use the cross-entropy loss for both classifiers

$$\mathcal{L}_{\text{class}}^{\text{drugs}} = \text{CE}\big(A_\phi^{\text{drug}}(z_i)\big), d_i\big) \quad \text{and} \quad \mathcal{L}_{\text{class}}^{\text{cov}} = \text{CE}\big(A_\phi^{\text{cov}}(z_i), c_i\big) \, .$$

Following the CPA implementation from Lotfollahi et al. (2021), we add a zero-centered gradient penalty to the loss function of the adversarial classifiers, to minimise

$$\mathcal{L}_{\text{pen}}^j = \frac{1}{k} \sum_k \big\| \partial_{z_i} A_\phi^j(z_i)_k \big\|_2^2 \, .$$

This gradient penalty was shown to make the discriminator more robust to noise and enable local convergence, when applied to generative adversarial networks (Mescheder et al., 2018). During training, we alternate update steps between the following competing objectives

$$\mathcal{L}_{\text{AE}}(\theta, \psi, \varphi | \phi) = \mathcal{L}_{\text{rec}}(\theta, \psi, \varphi) - \lambda_{\text{dis}} \sum_j \mathcal{L}_{\text{class}}^j(\theta \,|\, \phi) \quad \text{and}$$

$$\mathcal{L}_{\text{Adv}}(\phi \,|\, \theta) = \sum_j \mathcal{L}_{\text{class}}^j(\phi \,|\, \theta) + \lambda_{\text{pen}} \mathcal{L}_{\text{pen}}^j(\phi) \,,$$

where $\lambda_{\text{dis}}$ balances the importance of good reconstruction against the encoder $E_\theta$'s constraint to generate disentangled basal states $z_i$. The gradient penalty is weigthed with $\lambda_{\text{pen}}$.

## 4 Datasets and transfer learning

We use the sci-Plex3 (Srivatsan et al., 2020) and the L1000 (Subramanian et al., 2017) datasets for the main evaluation on single-cell data and pretraining on bulk experiments, respectively.

**Datasets**  The L1000 data contains about 1.3 million bulk RNA observations for 978 different genes. It includes measurements for almost 20k different drugs, some of which are FDA-approved, while others are synthetic compounds with no proven effect on any disease. Compared to scRNA-seq data, the L1000 data allows to explore a more diverse space of molecules which makes it ideal for pretraining.

The sci-Plex3 data is similar in size and contains measurements for 649,340 cells across 7561 drug-sensitive genes. On three human cancer cell lines—A549, MCF7, and K562—single-compound perturbations for 188 drugs at four different dosages— $10\,\text{nM}$, $100\,\text{nM}$, $1\,\mu\text{M}$, and $10\,\mu\text{M}$— are examined. Note that all cell lines and about 150 compounds overlap with the L1000 data. In addition, Srivatsan et al. (2020) assigned to all compounds one of 19 different modes of action (MoA), also called pathways. In contrast to the mechanism of action, which is related to the biochemical interaction between a molecule and a cell, the MoA describes the anatomical change that results from the exposure of cells to a drug-like molecule.

**Transfer learning**  As we train chemCPA with a Gaussian likelihood loss, the dataset was first normalised and then $\log(x + 1)$-transformed. Depending on the experiment, we further reduced the number of genes included in the single-cell data. In Section 5.1, we first subsetted both datasets to the same 977 genes which were identified via ensemble gene annotations. For the final experiment in Section 5.2, the considered gene set is increased as we hypothesize that more than the 977 L1000 genes are required to capture the variability within the single-cell data. To assess whether pretraining on L1000 is still beneficial in this scenario, we included 1023 highly variable genes (HVGs) from the sci-Plex3 data. That is, we consider 2000 genes in total.

For the extended gene set, chemCPA's input and output dimensions have to be adjusted to match the total number of 2000 genes. This is realized by adding two non-linear layers $h_{\text{enc}} : \mathbb{R}^{n_{\text{fine-tune}}} \to \mathbb{R}^{n_{\text{pretrain}}}$ and $h_{\text{dec}} : \mathbb{R}^{2n_{\text{pretrain}}} \to \mathbb{R}^{2n_{\text{fine-tune}}}$ to the autoencoder. The encoder becomes $\hat{E}_\theta = E_\theta\big(h_{\text{enc}}(x)\big)$ and the decoder becomes $\hat{D}_\psi = h_{\text{dec}}\big(D_\psi(z')\big)$. In our example, we have $n_{\text{pretrain}} = 977$ and $n_{\text{fine-tune}} = 2000$. We train all layers during fine-tuning, including the newly added ones. This architecture surgery differs from the procedure introduced in Lotfollahi et al. (2022), where individual neurons are added (instead of whole layers) and the transfer is performed on dataset labels (instead of gene sets).

## 5 Experiments

Our evaluation strategy tests chemCPA's ability to produce counterfactual predictions. To this end, it is important to measure both the predictive performance of a trained model as well as the degree to which the latent space components are disentangled.

**Counterfactual predictions**  To perform a counterfactual prediction, chemCPA first encodes an unperturbed control observations, a cell treated with dimethyl sulfoxide. The resulting basal state is then combined with the encoding of the desired drug and cell line, and chemCPA decodes the result. As we are free to choose any drug encoding, we refer to this process as counterfactual prediction.

Table 1: Comparison of multiple models on their performance on generalisation to unseen drug-covariate combinations for dosage values of $1\,\mu$M and $10\,\mu$M.

| Dose | Model | $\mathbb{E}[r^2]$ all | $\mathbb{E}[r^2]$ DEGs | Median $r^2$ all | Median $r^2$ DEGs |
|---|---|---|---|---|---|
| $1\,\mu$M | Baseline | 0.69 | 0.51 | 0.82 | 0.62 |
| | scGen | 0.73 | 0.59 | 0.77 | 0.68 |
| | CPA | 0.72 | 0.54 | **0.86** | 0.67 |
| | chemCPA | 0.74 | 0.60 | **0.86** | 0.66 |
| | chemCPA pretrained | **0.77** | **0.68** | 0.85 | **0.76** |
| $10\,\mu$M | Baseline | 0.50 | 0.29 | 0.48 | 0.12 |
| | scGen | 0.62 | 0.47 | 0.66 | 0.49 |
| | CPA | 0.54 | 0.34 | 0.52 | 0.26 |
| | chemCPA | 0.71 | 0.58 | 0.77 | 0.64 |
| | chemCPA pretrained | **0.76** | **0.68** | **0.82** | **0.79** |

**Evaluation strategy**  Throughout our experiments, we use the coefficient of determination $r^2$ as the main performance metric. This score is computed between the actual measurements and the counterfactual predictions on all genes and the 50 most differentially expressed genes (DEGs). It is necessary to consider all genes to evaluate the background and general decoder performance. However, the resulting $r^2$-scores can get inflated since most genes stay similar to their controls under perturbation. In contrast, the DEGs capture the differential signal which reflects a drug's effect. To further stress the importance to report both scores, note that the DEGs are unknown for unseen drugs as they depend on the drug and cell type. Hence, the combination is essential to gauge the accuracy of the model's predictions.

In order to classify the degree of disentanglement during evaluation, we train separate MLPs with four layers over 400 epochs and compute the prediction accuracy for drugs and covariates given the basal state. We consider the resulting accuracies as our disentanglement scores. An optimally disentangled model achieves scores that match the ratios of the most abundant drug and cell line, respectively. Since no model achieves perfect scores, we subset to models that are sufficiently disentangled. Throughout our experiments on the sci-Plex3 data, we set the thresholds for perturbation and cell line disentanglement to $< 10\%$ and $< 70\%$, respectively, while values of $3\%$ and $51\%$ are optimal. Note that poor disentanglement will automatically lead to low scores due to computing the test scores on the counterfactual predictions.

### 5.1    Comparing chemCPA against existing methods on unseen drug-covariate combinations

Before evaluating how well chemCPA can generalise to unseen drugs, we have to establish its competitive performance in a less ambitious setting. For this, we consider the scenario of generalisation to unobserved combinations of drugs and cell lines on the sci-Plex3 data and compare chemCPA against scGen (Lotfollahi et al., 2019) and CPA (Lotfollahi et al., 2021).

As scGen cannot distinguish between different dosage values, we perform two separate experiments for the second and highest dose values, $1\,\mu$M and $10\,\mu$M, respectively. Moreover, as both CPA and scGen require each individual component (drug $d$, cell line $c$) to be part of the training data $\mathcal{D}$, we create three distinct splits with only two of the three different drug set and cell line combinations being present during training, the third one being left for testing.

We choose to test nine different compounds: Dacinostat, Givinostat, Belinostat, Hesperadin, Quisi-nostat, Alvespimycin, Tanespimycin, TAK-901, and Flavopiridol. These drugs mostly belong to three MoA—epigenetic regulation, tyrosine kinase signalling, and cell cycle regulation—and were reported among the most effective drugs in the original publication (Srivatsan et al., 2020).

As discussed in the previous paragraph, we report mean and median $r^2$ values, which we averaged over the three splits and all drugs, for the sets of all genes and differentially expressed genes (DEGs). We consider a model that discards all perturbation information as our baseline. As a consequence, one can understand the improvement over the baseline as a result of the additional drug encoding. Moreover, we use the L1000 data for the pretraining of chemCPA and subset to the same 977 genes

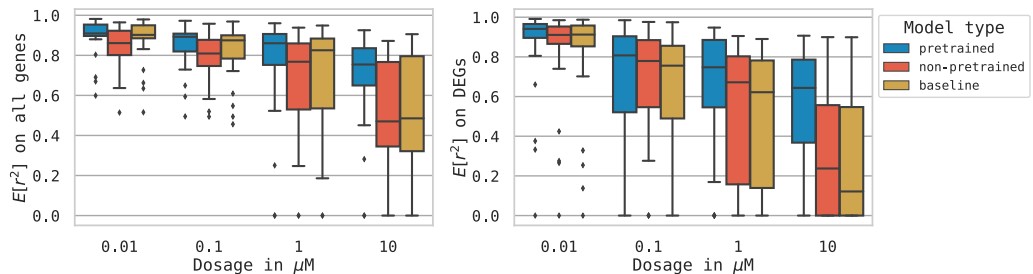

Figure 2: Performance of chemCPA on both the complete gene set (977 genes) and the compound specific DEGs (50 genes). In both cases, the pretrained model shows the best performance. At $10\,\mu$M on the DEGs, more than $50\%$ of the predictions have an $r^2$ score $> 0.6$ while the baseline's median is below $0.2$.

Table 2: Performance of chemCPA on the shared gene set. Since drug effects are stronger for high dosages, we present scores for a dosage value of $10\,\mu$M.

| Model | $\mathbb{E}[r^2]$ all | $\mathbb{E}[r^2]$ DEGs | Median $r^2$ all | Median $r^2$ DEGs |
|---|---|---|---|---|
| Baseline | 0.50 | 0.29 | 0.49 | 0.12 |
| chemCPA | 0.51 | 0.32 | 0.47 | 0.24 |
| chemCPA pretrained | **0.68** | **0.54** | **0.75** | **0.64** |

for the fine-tuning on sciPlex-3. This way, we are able to evaluate chemCPA in its pretrained and non-pretrained version.

We report the results of this experiment in Table 1. ChemCPA outperforms both scGen and CPA, demonstrating that the perturbation network together with pretraining leads to SOTA performance. Note also that the base version of chemCPA performs better than both CPA and scGen, indicating that the additional regularisation that comes from the perturbation networks $P_\varphi$ has beneficial effects on single-cell perturbation modelling.

To make a fair comparison, we optimised all CPA and chemCPA models identically and swept over the same set of hyperparameters (random, 10 samples). scGen was optimised with default parameters and an adjusted KL annealing scheme to match the set number of epochs. Note that both CPA and chemCPA can take control cells $x_i$ from all cell lines as input ({A549, K562, MCF7} $\rightarrow$ {A549, K562, MCF7}), while the cell line input for scGen has to match the test set (A549 $\rightarrow$ A549, K562 $\rightarrow$ K562, MCF7 $\rightarrow$ MCF7).

## 5.2 Using chemCPA to predict single-cell responses for unseen drugs

For the application of chemCPA to predict perturbation responses for unseen compounds, we use the same nine drugs from sci-Plex3 data as in Section 5.1. In addition to the shared gene set, we also consider an extended gene set, cf. Section 4. We include HVGs to account for the technological difference between bulk and single-cell and to capture the variance of single-cell data. To this end, the 977 genes present in both datasets are extended with 1023 HVGs of the sci-Plex3 data. Note that through this larger genes set, the 50 DEGs become a subset of the HVGs which makes it considerably more difficult for pretrained models to leverage learned bulk expressions directly.

**Shared gene set**    Table 2 shows the test performance of chemCPA, averaged over all drugs and the three cell lines, for the same gene set as used in Section 5.1. The pretrained chemCPA model consistently outperform the baseline and its base version.

The high baseline scores in Figure 2 shows that the drugs have almost no effect at low dosages. At high dosages, however, we see how chemCPA's predictions improve over the baseline. Looking at the prediction for all genes, the pretrained model has a significant advantage over its non-pretrained version. As expected, the performance is lower for the DEGs. Nevertheless, also in this scenario,

Table 3: We show the performance of chemCPA on the extended gene set. Since drug effects are stronger for high dosages, we present scores for a dosage value of $10\,\mu\mathrm{M}$.

| Model | $\mathbb{E}[r^2]$ all | $\mathbb{E}[r^2]$ DEGs | Median $r^2$ all | Median $r^2$ DEGs |
|---|---|---|---|---|
| Baseline | 0.37 | 0.19 | 0.16 | 0.00 |
| chemCPA | 0.46 | 0.22 | 0.35 | 0.00 |
| chemCPA pretrained | **0.69** | **0.47** | **0.79** | **0.62** |

chemCPA, especially the pretrained version, can explain gene expression values that must result from the drugs' influence.

In Figure 3, latent perturbations $z_d$ are visualised. Note that the difference between the lowest and highest doseage values results only from the non-linear dosage scaler $S$.

**Extended gene sets**  The extension to the larger gene set introduces a more difficult task for chemCPA. In Table 3, we show the same analysis as for the shared gene set. Again, the advantage of the pretraied chemCPA model translates to this scenario, while the base version is only slightly better than the baseline. A more comprehensive view for all dosages is shown in Figure 4, see also A.5 for results with different molecule encoders $G$.

This is a promising result, as it suggests that the transfer from abundant bulk RNA perturbation screens can be leveraged even in scenarios where the gene sets do not match. Crucially, this enables users to benefit from the proposed transfer learning and chemCPA's modelling capacity while simultaneously accounting for the special requirements of scRNA-seq data. In Figure 5 we show an explicit example for chemCPA's performance for two histone deacetylation drugs, see also Figure 13 in the appendix for more details.

Yet, for real applications it is essential to make limitations concerning the data and method transparent. To this end, we propose an uncertainty measure that addresses some of the limitations related to the generalisation to unseen compounds.

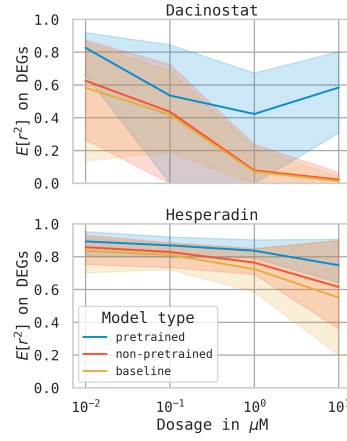

Figure 5: Perturbation prediction for Dacinostat and Hesperadin for chemCPA across all three cell-lines for the shared gene set.

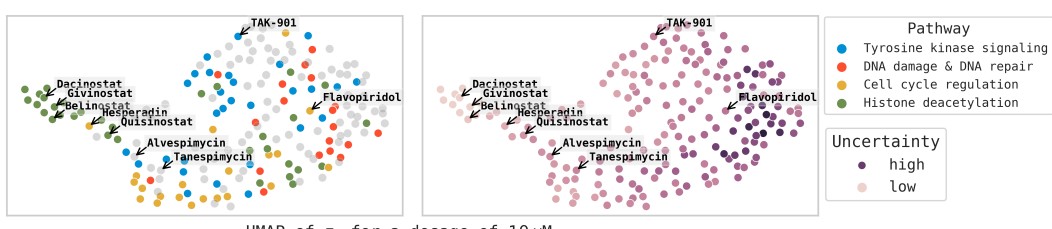

Figure 3: Illustration of the the scaled perturbation embedding $s \times z_d$ for $10\,\mu\mathrm{M}$. The left part illustrates how the perturbation embeddings $z_d$ are clustered according to some of the pathways. Most notably, the histone deacetylation drugs show a clear separation. Further context is provided by the uncertainty score on the right, showing regions of high and low confidence for the drug embeddings $z_d$. The nine test compounds are labeled.

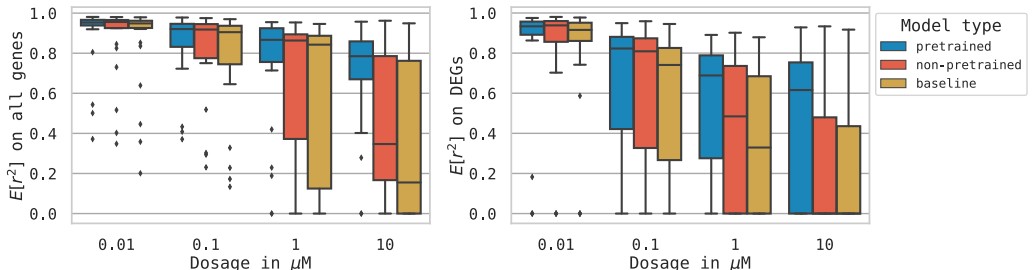

Figure 4: Performance of chemCPA model on the extended gene set, see also Figure 2. The pretrained model shows the best performance with the non-pretrained model failing to beat the baseline at lower dosages. At $10\,\mu$M on the DEGs, the pretrained model's median much higher that the baseline.

Table 4: Uncertainty score for all nine unseen drugs. The last column shows the improvement $\Delta r^2$ of chemCPA over the baseline. The number of considered neighbours was nine for all drugs.

| Drug | Uncertainty score $u$ | $\Delta r^2$ on DEGs |
|---|---|---|
| Dacinostat | 0.570 | 0.00 |
| Givinostat | 0.660 | 0.00 |
| Belinostat | 0.623 | 0.12 |
| Hesperadin | 0.197 | 0.40 |
| Quisinostat | 0.052 | 0.65 |
| Alvespimycin | 0.058 | 0.74 |
| Tanespimycin | 0.092 | 0.75 |
| TAK-901 | 0.049 | 0.88 |
| Flavopiridol | 0.011 | 0.99 |

### 5.3 Measure uncertainty on the drug embedding

Generalisation can only be achieved within the limits of the dataset a model is trained on. For the sci-Plex3 data, less than $20\%$ of the drug-dose combinations deviate from the controls' phenotypes by more than $35\%$ in its $r^2$ scores. In additon, we know from the original sci-Plex3 publication that only drugs from a few pathways—tyrosine kinase signaling, DNA damage and repair, cell cycle regulation, and epigenetic regulation—show a clear effect. We assume that this technological noise is the reason why the non-pretrained chemCPA version struggles to outperform the baseline on the extended gene set, whereas the pretrained model is more robust. These data challenges are also reflected in the left part of Figure 3 as we would expect chemCPA's perturbation latent space to cluster according to the drugs' MoA, similar to the cluster of histone deacetylation drugs.

We found that an imperfect clustering often correlates with high baseline scores and, as a result of that, chemCPA not being able to identify distinct perturbations, see Figure 15 in the appendix. To make the generalisation ability more transparent, we employ a measure of uncertainty. A good indicator for chemCPA's ability to generalise is the MoA prediction from the KNN-graph of the perturbation embedding space. We further combine this measure with the average distance to neighbouring drugs as we recognise that larger distances indicate a distinct perturbation:

$$u_i = \sum_{j \in \mathcal{N}_i} \frac{1}{\log\left(d(i,j)\right)} \times H(X) \quad,$$

where $d$ is the Euclidean distance, $H$ is the Shannon entropy, and $X$ the normalised pathway prediction deduced from the neighbours $\mathcal{N}_i$ of drug $i$. This uncertainty measure combines two things: First, chemCPA's confidence on the drug's MoA, measured by $H$, and second, whether chemCPA expects the drug to have a distinct perturbation effect on the cell, measured by the inverse distance.

We report an analysis of the uncertainty for the chemCPA model in Figure 3 and Table 4. A plot that shows chemCPA's performance and uncertainty for all compounds is part of the appendix A.6. The results show that the uncertainty score $u$ for unseen drugs correlates well with the accurate prediction

of perturbed cells. This illustrates how chemCPA's compositional latent space can be leveraged for additional insights in order to evaluate its generalisation ability.

## 6  Conclusion

In this paper, we introduced chemCPA, a model for predicting cellular gene expression responses for unseen drug perturbations by encoding the drugs' molecular structures. We showed how chemCPA outperforms CPA and scGen on shared tasks, while generalising over existing methods by being applicable to the novel task of generalising to unseen drugs. Applied to single-cell data, we demonstrated how pretraining on bulk HTSs improves chemCPA's generalisation performance. This applies even when the gene set of the single-cell dataset differs from the genes of the pretraining bulk RNA HTS dataset. We further provided an uncertainty measurement that correlates well with the chemCPA's generalisation ability to unseen drugs. Taken all results together, we are confident that chemCPA will benefit from higher-quality scRNA-seq HTSs in the future and can become a powerful aid in the drug screening and drug discovery process.

## Acknowledgments and Disclosure of Funding

LH is thankful for valuable feedback from Fabiola Curion and Carlo De Donno. LH is supported by the Helmholtz Association under the joint research school "Munich School for Data Science - MUDS". ML is grateful for financial support from the Joachim Herz Stiftung. NK and FJT ackowledge support by Helmholtz Association's Initiative and Networking Fund through Helmholtz AI (ZT-I-PF-5-01). FJT further acknowledges support by the BMBF (01IS18053A). FJT consults for Immunai Inc., Singularity Bio B.V., CytoReason Ltd, and Omniscope Ltd, and has ownership interest in Dermagnostix GmbH and Cellarity. This work was supported by the German Federal Ministry of Education and Research (BMBF) under Grant No. 01IS18036B.

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
