# A  Appendix

## A.1  Benchmarking drug molecule encoders

Enabled by the flexibility of the molecule encoder $G$, we investigated what impact the architecture choice has on the performance of chemCPA. For this, we compared multiple pretrained graph-based models whose weights were frozen during the training. Next to predefined RDKit fingerprints (which are non-differentiable, and hence not trainable), we included a GCN, MPNN, weave model, GROVER model, and a JT-VAE.

Table 5 summarises the results of this experiment on the L1000 dataset. The weave model performs much worse than the others, achieving an $r^2$-score of only $65 \pm 8$ on DEGs. While the GCN disentangles well, it is outperformed by the JT-VAE and GROVER models. All experiments in the main text were ran across all three best performing models (GROVER, JT-VAE and RDKit), however due to space constraints we only report RDKit results in the main text.

Table 5: Summary of chemCPA on the L1000 dataset for different molecule encoders $G$. All models were trained on the same random split. Reported are the overall disentanglement scores (drug and cell line) and the $r^2$-scores on the test set.

| Model $G$ | Drug | Cell line | Mean $r^2$ all | Mean $r^2$ DEGs |
|---|---|---|---|---|
| GCN | $\mathbf{0.08 \pm 0.03}$ | $\mathbf{0.17 \pm 0.01}$ | $0.92 \pm 0.01$ | $0.81 \pm 0.05$ |
| MPNN | $0.10 \pm 0.03$ | $0.28 \pm 0.07$ | $0.92 \pm 0.01$ | $0.82 \pm 0.03$ |
| GROVER | $0.09 \pm 0.03$ | $0.19 \pm 0.04$ | $0.93 \pm 0.01$ | $\mathbf{0.87 \pm 0.01}$ |
| JT-VAE | $\mathbf{0.08 \pm 0.02}$ | $0.20 \pm 0.04$ | $0.93 \pm 0.01$ | $\mathbf{0.87 \pm 0.01}$ |
| RDKit | $0.10 \pm 0.04$ | $0.29 \pm 0.13$ | $0.93 \pm 0.01$ | $0.85 \pm 0.03$ |
| weave | $0.10 \pm 0.03$ | $0.29 \pm 0.09$ | $0.89 \pm 0.02$ | $0.65 \pm 0.08$ |

Table 6 shows the test performance of chemCPA for the nine unseen drugs across the three cell lines in the transfer learning scenario with identical genes. This is the same experiment as Table 2, but evaluated across more embedding models. The fine-tuned chemCPA models for GROVER and RDKit consistently outperform the baseline and their non-pretrained version with RDKit achieving the highest median score on DEGs. Interestingly, the fine-tuned JT-VAE model is better than the baseline and other non-pretained chemCPA models but worse than its own non-pretrained version.

Table 6: Performance of pretrained and non-pretrained chemCPA models across the three versions of the molecule encoder $G$, for the L1000 to SciPlex3 transfer learning experiment with shared gene sets. Since drug effects are stronger for high dosages, the scores are evaluated at a dosage value of $10\,\mu$M.

| Model $G$ | Type | Mean $r^2$ all | Mean $r^2$ DEGs | Median $r^2$ all | Median $r^2$ DEGs |
|---|---|---|---|---|---|
|  | baseline | 0.50 | 0.29 | 0.49 | 0.12 |
| GROVER | non-pretrained | 0.52 | 0.32 | 0.51 | 0.18 |
|  | pretrained | 0.63 | 0.47 | 0.70 | 0.49 |
| JT-VAE | non-pretrained | 0.60 | 0.39 | 0.68 | 0.42 |
|  | pretrained | 0.55 | 0.35 | 0.55 | 0.28 |
| RDKit | non-pretrained | 0.51 | 0.32 | 0.47 | 0.24 |
|  | pretrained | $\mathbf{0.68}$ | $\mathbf{0.54}$ | $\mathbf{0.75}$ | $\mathbf{0.64}$ |

In Table 7, we show the same experiment as in Table 3. Again, the pretrained chemCPA model with an RDKit molecule encoder $G$ perform best. We believe that this can be attributed to two things. First, the sci-Plex3 data is the first of its kind, and technological noise is still an issue. When evaluated over the whole training set, the baseline achieves $r^2$-scores higher than $65\%$ for more than $96\%$ of the observations. This sparsity might hinder the more complex perturbation networks $P_\varphi$, which are based on GROVER and JT-VAE, from finding good perturbation representations. We suspect that the same reason also explains the bad performance of non-pretrained models as these are

more susceptible to noise, whereas the fine-tuned models are more robust. Second, the pretrained embedding $h$ that result from RDKit identifies the histone deacetylation drugs as a distinct cluster, see Figure 10. Since these compounds show the strongest effect in the sci-Plex3 data, the inductive bias from RDKit give an explanation for the favourable generalisation performance.

Table 7: Performance of pretrained and non-pretrained chemCPA models across the three versions of the molecule encoder $G$ on the extended gene set. Since drug effects are stronger for high dosages, the scores are evaluated at a dosage value of $10\,\mu$M.

| Model $G$ | Type | Mean $r^2$ all | Mean $r^2$ DEGs | Median $r^2$ all | Median $r^2$ DEGs |
|---|---|---|---|---|---|
|  | baseline | 0.37 | 0.19 | 0.16 | 0.00 |
| GROVER | non-pretrained | 0.41 | 0.22 | 0.28 | 0.00 |
|  | pretrained | 0.59 | 0.36 | 0.75 | 0.45 |
| JT-VAE | non-pretrained | 0.40 | 0.22 | 0.20 | 0.00 |
|  | pretrained | 0.51 | 0.24 | 0.51 | 0.00 |
| RDKit | non-pretrained | 0.46 | 0.22 | 0.35 | 0.00 |
|  | pretrained | **0.69** | **0.47** | **0.79** | **0.62** |

## A.2 Attribute embedding

Table 8: Details on pretrained models for the molecule encoder $G$.

| Molecule encoder $G$ | Embedding dim $h_{\text{drug}}$ | Pretrained |
|---|---|---|
| RDKit | 200 | – |
| GROVER | 3400 | authors |
| JT-VAE | 56 | ZINC, L1000, sci-Plex3 |
| GCN | 128 | PCBA |
| MPNN | 128 | PCBA |
| weave | 128 | PCBA |

## A.3 Counterfactual prediction

1. To compute counterfactual predictions, we obtain basal states $z_i$ for all control observations present in the test set. For each combination of drug, dose, and cell line in the test set, we compute the latent attribute state $z_{\text{attribute}}$ and combine it with all $z_i$. Subsequently, we compute the mean per gene across all predictions and likewise for the real measurements. As a result, we obtain two $n$ dimensional vectors, where $n$ is the number of genes (977 or 2000), for which we compute the $r^2$ score. Taken together, we get one score per combination.

## A.4 Additional information on the L1000 experiment

1. For infos on the RDKit sweep and resulting best run, see Table 9 and Table 10.
2. Architectures for best configuration of the perturbation networks $P_\varphi$ and adversary classifiers are presented in Table 11.
3. For details on the performance of the best runs, see Table 12.

Table 9: Fixed Parameters for the RDKit sweep in the L1000 dataset.

| Parameter | Value |
|---|---|
| num_epochs | 1500 |
| dataset_type | lincs |
| decoder_activation | linear |
| model | rdkit |

Table 10: Random parameters for the RDKit sweep in the L1000 dataset.

| Parameter | Type | Values | Best config |
|---|---|---|---|
| samples | fixed | 25 | NaN |
| dim | choice | {64, 32} | 32 |
| dosers_width | choice | {64, 256, 128, 512} | 64 |
| dosers_depth | choice | {1, 2, 3} | 1 |
| dosers_lr | loguniform | $[1 \times 10^{-4}, 1 \times 10^{-2}]$ | $5.61 \times 10^{-4}$ |
| dosers_wd | loguniform | $[1 \times 10^{-8}, 1 \times 10^{-5}]$ | $1.33 \times 10^{-7}$ |
| autoencoder_width | choice | {128, 256, 512} | 256 |
| autoencoder_depth | choice | {3, 4, 5} | 4 |
| autoencoder_lr | loguniform | $[1 \times 10^{-4}, 1 \times 10^{-2}]$ | $1.12 \times 10^{-3}$ |
| autoencoder_wd | loguniform | $[1 \times 10^{-8}, 1 \times 10^{-5}]$ | $3.75 \times 10^{-7}$ |
| adversary_width | choice | {64, 256, 128} | 128 |
| adversary_depth | choice | {2, 3, 4} | 3 |
| adversary_lr | loguniform | $[5 \times 10^{-5}, 1 \times 10^{-2}]$ | $8.06 \times 10^{-4}$ |
| adversary_wd | loguniform | $[1 \times 10^{-8}, 1 \times 10^{-3}]$ | $4.0 \times 10^{-6}$ |
| adversary_steps | choice | {2, 3} | 2 |
| reg_adversary | loguniform | [5, 100] | 24.1 |
| penalty_adversary | loguniform | [1, 10] | 3.35 |
| batch_size | choice | {32, 64, 128} | 128 |
| step_size_lr | choice | {200, 50, 100} | 100 |
| embedding_encoder_width | choice | {128, 256, 512} | 128 |
| embedding_encoder_depth | choice | {2, 3, 4} | 3 |

Table 11: Presented are the best configurations per molecule encoder from 18 random hyperparamter samples similar to the one presented in Table 10.

| Parameter | GROVER | MPNN | RDKit |
|---|---|---|---|
| dosers_width | 512 | 64 | 64 |
| dosers_depth | 2 | 2 | 3 |
| dosers_lr | $5.61 \times 10^{-4}$ | $1.58 \times 10^{-3}$ | $1.12 \times 10^{-3}$ |
| dosers_wd | $1.33 \times 10^{-7}$ | $6.25 \times 10^{-7}$ | $3.75 \times 10^{-7}$ |
| embedding_encoder_width | 512 | 128 | 128 |
| embedding_encoder_depth | 3 | 4 | 4 |

| Parameter | weave | JT-VAE | GCN |
|---|---|---|---|
| dosers_width | 512 | 64 | 512 |
| dosers_depth | 2 | 2 | 2 |
| dosers_lr | $1.12 \times 10^{-3}$ | $2.05 \times 10^{-4}$ | $2.05 \times 10^{-4}$ |
| dosers_wd | $2.94 \times 10^{-8}$ | $2.94 \times 10^{-8}$ | $1.33 \times 10^{-6}$ |
| embedding_encoder_width | 128 | 256 | 128 |
| embedding_encoder_depth | 3 | 4 | 3 |

## A.5 Additional information on the sci-Plex3 experiments

1. The optimisation was performed similarly to the presented sweeps in Table 10 and Table 11 for the perturbation network and adversary parameters for 10 samples each per category.

2. Boxplot results for RDKit, see Figures 6 and 8, and JT-VAE, see Figures 7 and 9.

3. Paired t-tests were performed for both settings, see Table 14 for the shared gene set and Table 14 for the extended gene set.

4. More examples on the performance with respect to specific drugs are presented in Figure 11, Figure 12, Figure 13, and Figure 14.

5. The Drug embedding that results from RDKit is shown in

Table 12: Performance of the best runs on L1000 for different molecule encoders $G$

| Model $G$ | Drug | Cell line | Mean $r^2$ all | Mean $r^2$ DEGs | Mean $r^2$ DEGs [val] |
|---|---|---|---|---|---|
| GCN | 0.11 | 0.16 | 0.92 | 0.84 | 0.83 |
| MPNN | 0.07 | 0.24 | 0.94 | 0.87 | 0.84 |
| GROVER | 0.07 | 0.16 | 0.94 | 0.88 | 0.86 |
| JT-VAE | 0.06 | 0.15 | 0.94 | 0.88 | 0.85 |
| RDKit | 0.08 | 0.15 | 0.93 | 0.86 | 0.85 |
| weave | 0.09 | 0.20 | 0.91 | 0.74 | 0.72 |

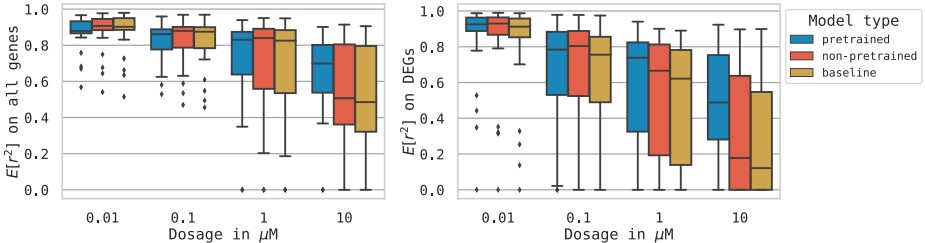

Figure 6: Performance of the pretrained and non-pretrained chemCPA model using GROVER. Comparisons against the baseline are done on both the complete gene set (977 genes) and the compound specific DEGs (50 genes).

## A.6 Additional information on the uncertainty score

1. The uncertainty computation for the chemCPA model with an RDKit molecule embedding for the shared gene setting is shown in Figure 15.

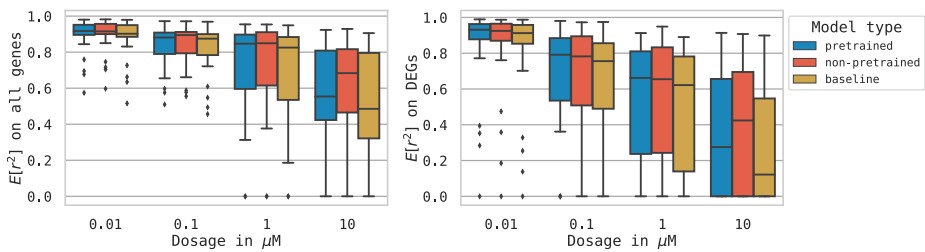

Figure 7: Performance of the pretrained and non-pretrained chemCPA model using JT-VAE. Comparisons against the baseline are done on both the complete gene set (977 genes) and the compound specific DEGs (50 genes).

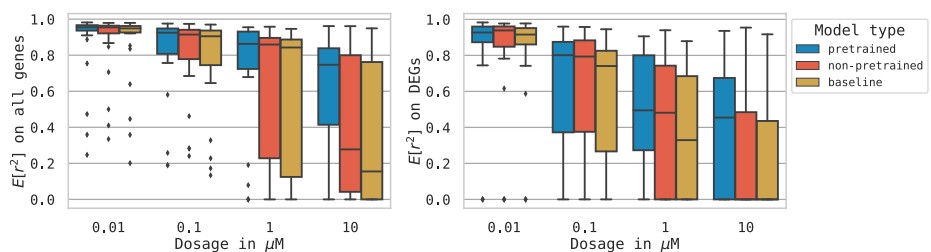

Figure 8: Performance of the pretrained and non-pretrained chemCPA model on the extended gene set using GROVER, see also Figure 6.

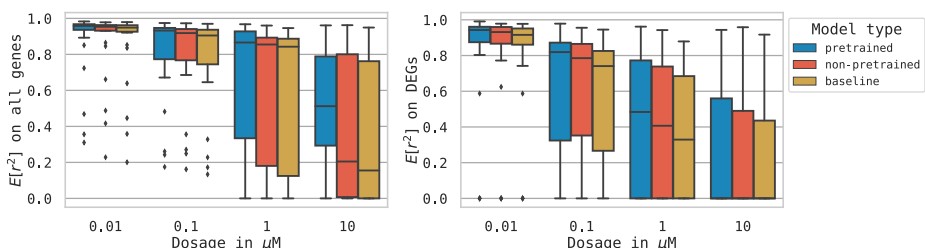

Figure 9: Performance of the pretrained and non-pretrained chemCPA model on the extended gene set using JT-VAE, see also Figure 7.

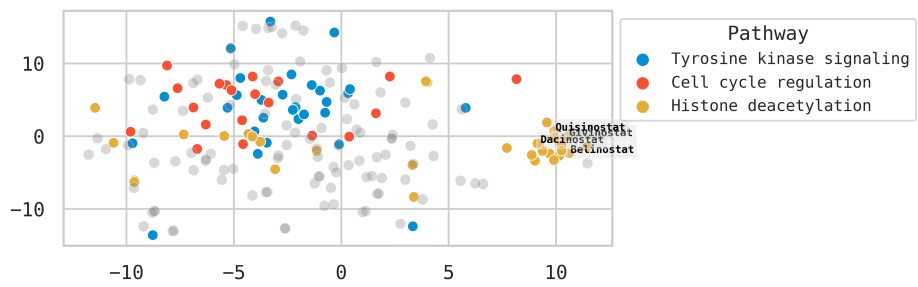

Figure 10: TSNE embedding based on the RDKit features of the 188 drugs.

Table 13: Significance test for a dosage of $10\,\mu$M on the shared gene set using the paired t-test.

| Model $G$ | Against | Gene set | p-value |
|---|---|---|---|
| rdkit | baseline | all genes | 0.0002 |
| rdkit | baseline | DEGs | 0.0001 |
| rdkit | non-pretrained | all genes | 0.0001 |
| rdkit | non-pretrained | DEGs | 0.0003 |
| grover | baseline | all genes | 0.0008 |
| grover | baseline | DEGs | 0.0002 |
| grover | non-pretrained | all genes | 0.0023 |
| grover | non-pretrained | DEGs | 0.0022 |
| jtvae | baseline | all genes | 0.0002 |
| jtvae | baseline | DEGs | 0.0004 |
| jtvae | non-pretrained | all genes | 0.0141 |
| jtvae | non-pretrained | DEGs | 0.0528 |

Table 14: Significance test for a dosage of $10\,\mu$M on the extended gene set using the paired t-test.

| Model $G$ | Against | Gene set | p-value |
|---|---|---|---|
| rdkit | baseline | all genes | 0.0001 |
| rdkit | baseline | DEGs | 0.0004 |
| rdkit | non-pretrained | all genes | 0.0003 |
| rdkit | non-pretrained | DEGs | 0.0020 |
| grover | baseline | all genes | 0.0009 |
| grover | baseline | DEGs | 0.0038 |
| grover | non-pretrained | all genes | 0.0029 |
| grover | non-pretrained | DEGs | 0.0165 |
| jtvae | baseline | all genes | 0.0005 |
| jtvae | baseline | DEGs | 0.0024 |
| jtvae | non-pretrained | all genes | 0.0026 |
| jtvae | non-pretrained | DEGs | 0.0721 |

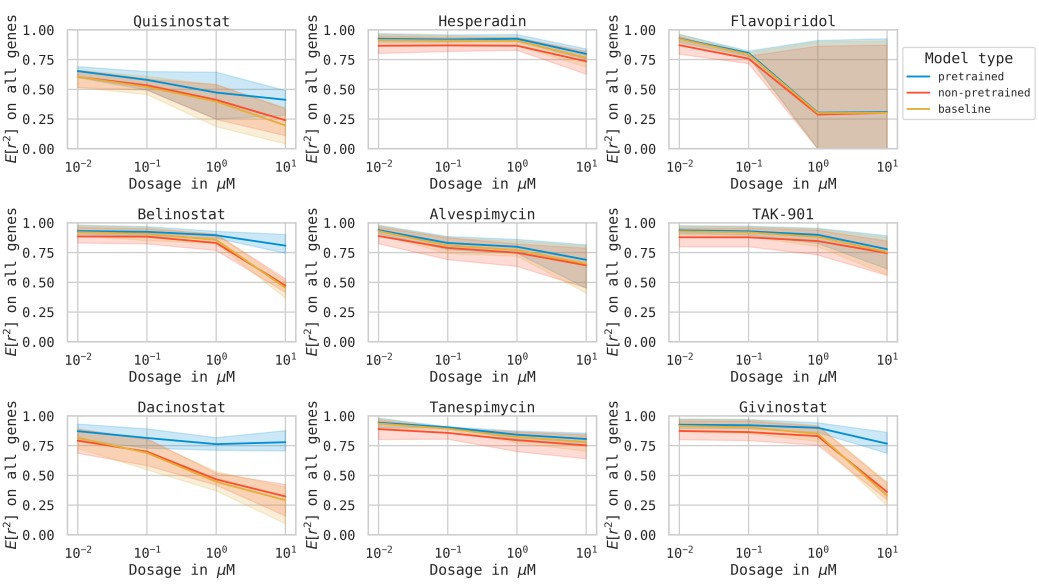

Figure 11: Drug-wise comparison between the baseline, pretrained and non-pretrained models using RDKit for all nine drugs in the test set considering all genes for the shared gene set.

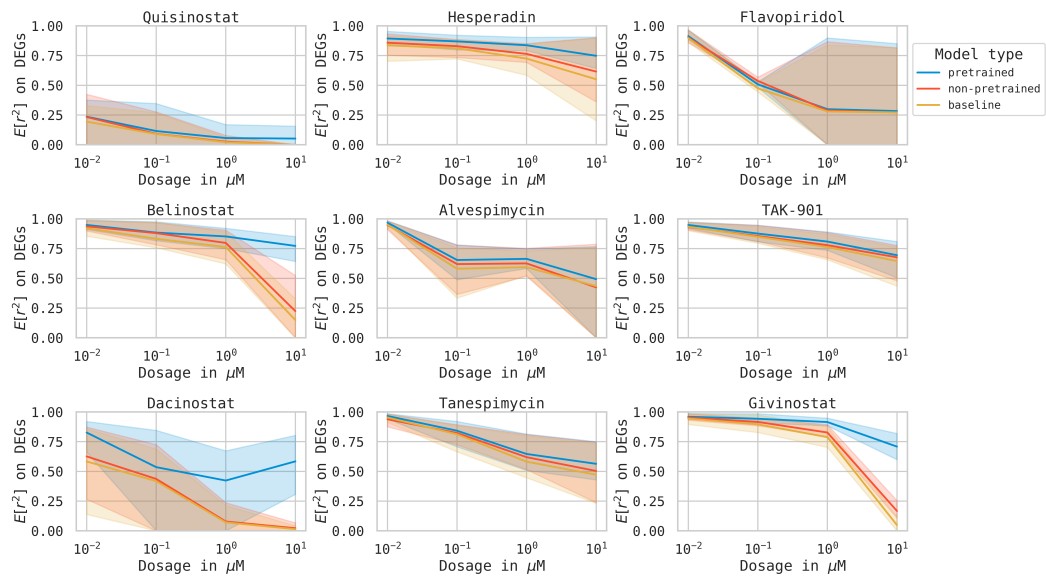

Figure 12: Drug-wise comparison between the baseline, pretrained and non-pretrained models using RDKit for all nine drugs in the test set considering the DEGs for the shared gene set.

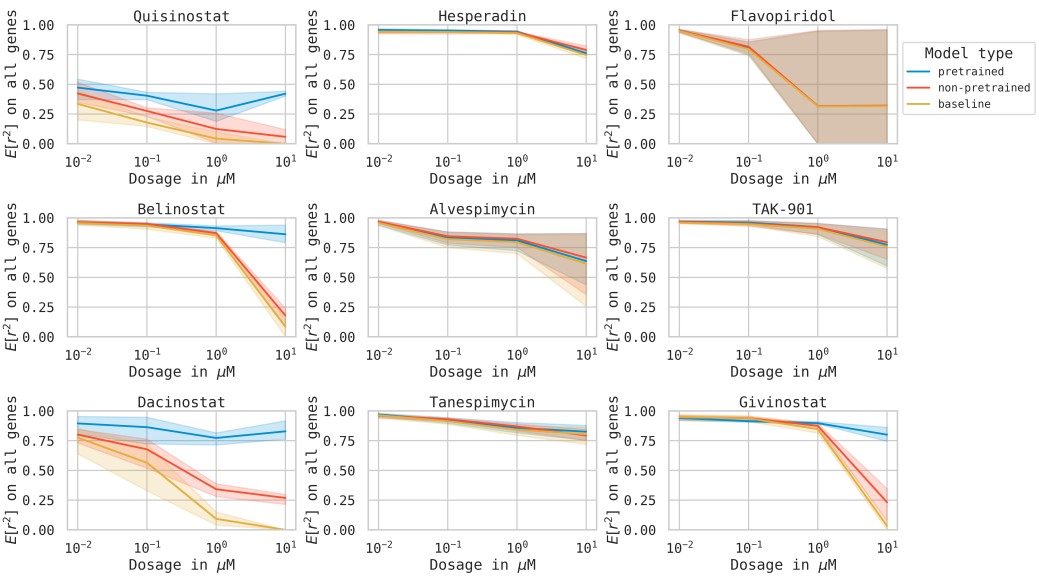

Figure 13: Drug-wise comparison between the baseline, pretrained and non-pretrained models using RDKit for all nine drugs in the test set considering all genes for the extended gene set.

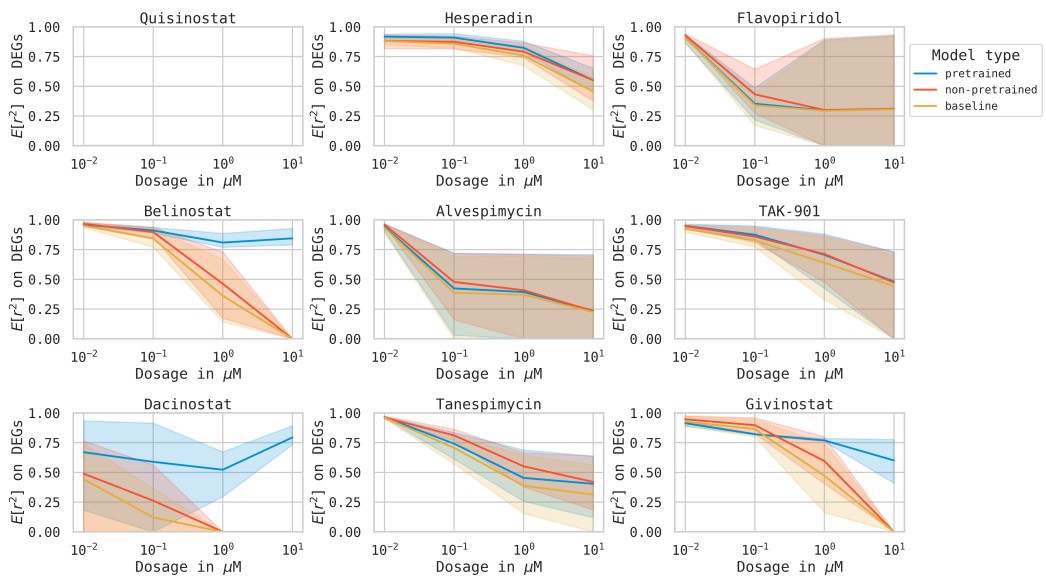

Figure 14: Drug-wise comparison between the baseline, pretrained and non-pretrained models using RDKit for all nine drugs in the test set considering the DEGs for the extended gene set.

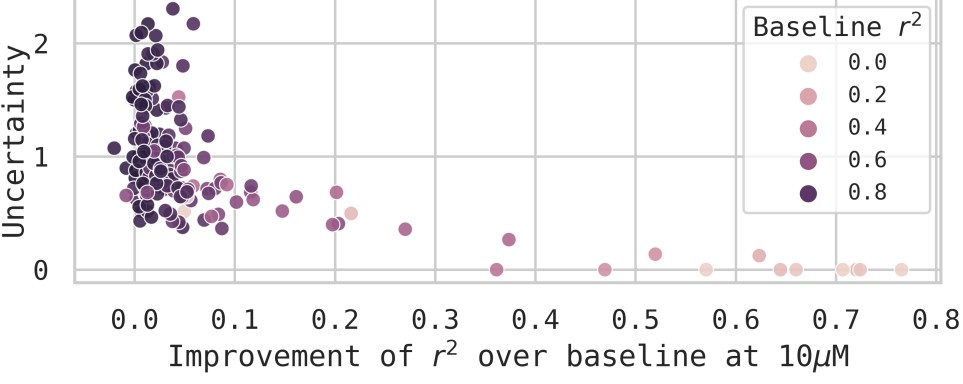

Figure 15: Uncertainty score for chemCPA's prediction on the perturbation embedding in relation to the model's improvement over the baseline score, measured in $r^2$.