# OpenReview forum: "Predicting Cellular Responses to Novel Drug Perturbations at a Single-Cell Resolution"
_NeurIPS.cc/2022/Conference — NeurIPS 2022 Accept_

### Official Review · Reviewer_32go · 2022-07-05

**Rating:** 8
**Confidence:** 4
**Soundness:** 3 good
**Presentation:** 3 good
**Contribution:** 3 good

**Summary:**

The paper suggests a new model (coined chemCPA) that predicts the transcriptomic perturbation of drugs on specific cell lines.
One of the highlights is that chemPCA uses some transfer learning approach to predict those perturbation at single cell level (for data is generally noisy and not widely available) while leveraging the vast amount of available bulk RNA-seq screens, even in the setting where the available set of genes are different between the source and the target domains.
Building on the previously published CPA model, chemPCA relies on an  a similar autoencoder architecture but uses existing pre-trained structure embeddings of the drugs to allow generalization to unseen compounds (or unseen combinations of seen compounds).
After introducing and briefly justifying the chemCPA architecture, the authors evaluate the performance of their method in an extensive setting, benchmarking first the choice of various chemical encoders, then validating their transfer learning (from bulk RNA-seq to the scRNA-seq domain) and finally the generalization performance to unseen drug compounds.
Last, the authors suggest an uncertainty measure on the drug embedding which experimentally seems to correlate well with how well chemPCA generalizes to unseen compounds.


09/08/2022 : after carefully reviewing the author's response to all reviewers, as well as the amended manuscript, I am updating my rating of the presentation from 2 to 3 and my overall rating of the paper from 5 to 8.

**Questions:**

In addition to comments on the points I raised in the previous section, here is a list of more specific questions I would love to see the authors answer:
 1) While it may be a very reasonable idea, I am under the impression that assuming that basal states and cell lines can be completely disentangled is somewhat of a strong statement. Do you have any argument or insight that support this choice of modelling? Or is it an approximation that is necessary to make the problem tractable? Some further comments would be very appreciated (and would, I think strengthen the manuscript if included).
 2) If I understood correctly (but I admittedly did not dig very deep into [Lotfollahi et al. '21]), the current setting is limited to predicting the perturbation induced in a fixed, known in advance, set of cell lines. Do you think this work could be extended to generalize to different cell lines, or, perhaps more interestingly to predicting perturbations in cells that are not coming from a cell line (e.g. from a biopsy)? Do you have insights on whether it would be completely unrealistic to use a vector of gene expression levels before perturbation as an input rather than an encoding of the cell line?
 3) A Gaussian likelihood reconstruction loss is used (and the data is log-normalized and standardized as a preprocessing step). Why not using a (Zero-Inflated?)Negative Binomial model that should be a better fit to such count data? Is it something that you tried or considered?
 4) There is no discussion on the choice of this additive model for the perturbations. I could imagine that the non-linearities induced by the encoder and decoder architectures mean that such an additive model may very well not be limiting the expressivity of chemCPA. It would be nice to have some comments about it, though.
 5) Related to the last question but more specific, reducing the dosage attributes to a single scalar value applied to the drug embedding vector intuitively feels like a very strong underlying assumption. Do you have any comment on that?
 6) A gradient penalty term (inspired from the Improved WGAN paper) is added to the adversarial loss. First, in [Gulrajani et al. '17], it was penalizing the gradients to have norm close to 1 (not a penalty on the norm directly). Second, having gradient norms equal to 1 was a proxy to having a 1-Lipschitz critic / discriminator because that is what made optimizing the WGAN objective equivalent to minimizing the Wasserstein distance between real and generated samples. Here, I don't see the connection at all. Any comment on the choice of this regularizer?


**Limitations:**

No discussion about the societal impact was included in the manuscript. But like the authors, I also do not foresee any impact so I don't see it as a problem.
In general, as written in the previous sections, my sense is that the manuscript does not comment enough on the modelling choices that were made. As such, I find the limitations of their work not to be very carefully addressed. Please refer to the other sections of this review for more details.

**Strengths And Weaknesses:**

Given the amount of efforts and resources spent on screening, in vitro, chemical compounds in the field of drug design, developing efficient and reliable in silico models that are able to predict the perturbations induced by such compounds on a transcriptomic level is of utmost importance.
While this problem has been the object of a lot of research over the last decades, multiple limitations still makes it an open problem.
Although I am not an expert of this subfield, I think that addressing specifically the following questions: 1) how to generalize predictions to unseen drugs (or combination of) and 2) if it is possible to transfer information from the very abundant and less noisy (but "lower resolution") bulk HTS data to the emerging (hence sparsely available for the moment) single cell HTS, is key to the development of the field.
While chemPCA admittedly inspires from already existing ideas, their combination enables novel applications and use cases. As such, I consider the novelty of this work to be satisfactory and its significance high.

I have some concerns about the clarity of the manuscript.
The experimental results of this paper are more extensive than the average NeurIPS submission with settings that require a lot of explanations.
I have the impression that this work would benefit from a longer format to improve (e.g. a bioinformatics or computational biology journal).
For a ML venue such as NeurIPS, I would have appreciated a bit more insights on some of the choices / assumptions made (see the Questions below for more details).
Some important details seem to be completely eluded. For instance, there is a mention of an "architecture surgery" to match the larger dimension of the extended set of genes of the target domain.
However, no real explanation is provided as to how this is executed. Is it related to the scArches paper [Lotfollahi et al. '22]? If yes an explicit reference to it would be welcome and the manuscript would benefit from more explanation (at least in the appendices).
Also, on a more technical level, many of the abbreviations (CPA, MoA, DMSO etc.) used are not defined properly (not defined at all, or after they were first used). It is a simple detail but fixing this would improve the clarity.
In Figure 3, the test compounds are labelled, but it is very difficult to know which label corresponds to which point. That makes it hard to even see if the test compounds are actually clustered at all with other drugs affecting the same pathway.

I also have concerns about the experimental results.
I find the setting and design of the experiments to be very thorough and interesting.
My main concern is that it is mostly a very elaborate ablation study where the impact of the choice of the drug encoder, the impact of pretraining are studied (also in terms of generalization to unseen drugs).
However, there is no comparison to any competing method or baseline (other than using the unperturbed measurements).
I understand that chemCPA has a quite unique positioning and that there are likely no other methods that predict perturbations of unseen drugs on single cells.
However, I think it would be relevant to test chemCPA on less ambitious settings where it can be compared to existing methods (e.g. predicting the perturbation of unseen pairs of cell lines and seen drugs).
Significantly improving over the SotA on such tasks would not be necessary, but showing that the results are not bad in such settings would, in my opinion, make the manuscript stronger.

I am well aware that my remarks go in the direction of adding even more content to an already very dense paper. Maybe some of the experimental details and results could be relegated to the appendices.

Overall, I have no doubt that the work underlying this submission is novel and significant enough to be the object of a relatively high impact publication. However, I think that in its current form, the paper is not reflecting the high quality and potential of this work.
If the authors answer my concerns convincingly and have a plan to update their manuscript accordingly, I could very well be convinced to improve my rating.

---

> ### Author Response · Authors · 2022-08-02
> **Addressing concerns about experimental results with an additional experiment, including additional comments on modelling assumptions**
>
> We thank the reviewer for their accurate summary, detailed evaluation, and multiple suggestions for our manuscript. We provide answers to the raised questions down below. For the manuscript, we added a new section that builds on the results of the experimental comparison presented in the general reply.
>
> ### Model assumptions
>
> #### Reasoning about chemCPA's design added
>
> >*Basal states and cell lines can be completely disentangled is somewhat of a strong statement*
>
> - The idea of separation into basal, drug, and cell states is indeed an approximation to make the problem tractable. The separation into covariates for which labels are available allows us to attribute perturbation effects to specific sources, e.g., drug or cell line. In that sense, the disentanglement has relevance for the biological application and potential downstream analyses [L140-143].
>
> >*Discussion on the choice of this additive model*
>
> - You are right, the non-linearities in the encoder, decoder, and perturbation network allow chemCPA to use such relatively simple arithmetics in the latent space while maintaining complexity [L121-126].
>
> >*Reducing the dosage attributes to a single scalar*
>
> - Note that the dosage scaler $\hat s$ is a function of the drug embedding $h_d$ and the applied dose $s$. One reason that we reduced it again to a single scalar is to compute drug-response curves easily. This way of encoding the dose also matches the idea that $z_d$ encodes the drug's effect. We are happy to provide such drug-response curves for the selected nine unseen drugs in the revised manuscript if interested [L112-114].
>
> >*gradient penalty term*
>
> - We added the gradient penalty inspired by the [original CPA implementation](https://github.com/facebookresearch/CPA). During hyperparameter sweeps we observed the penalty term improving latent space disentanglement. Somewhat similar techniques are used for GANs, but the previous reference was misleading, and we've updated our manuscript [L145].
>
> ### Transfer learning
>
> #### Details on architecture surgery added
>
> >*"architecture surgery" related to the scArches*
>
> - We agree that we did not describe chemCPA's architecture surgery in detail and that these should be included as it enables to leverage pretrained models from different gene sets. We added more details in the manuscript [L175-180]. To clarify, this is not related to the proposed strategy of scArches. While scArches add so-called anchors for new batches, we extend the gene set through adding an additional layer to both the encoder and decoder.
>
> ### Further extension
>
> #### chemCPA's aspect of transfer learning could enable generalisation to new cell types
>
> >*could work be extended to generalize to different cell lines*
>
> - ChemCPA already disentangles on the cell lines. Hence, generalisation to new drug-covariate combinations is possible and we demonstrate this with the new experiment [Sec.5.1]. Generalization to unseen cell lines ($c\notin\mathcal{D}$) requires a cell line embedding. This is in principle also possible, yes. In this work, we focused on the drug aspect which already requires transfer learning. Given an informative cell line embedding, it would be straightforward to extend chemCPA similarly as we did for molecules. To properly analyse an appropriate transfer learning and amortisation scheme would require a different dataset, however.
> - While it would be possible to encode any gene expression vector, we do not expect chemCPA to be reliable in this setting. The reason for this is that it would violate the assumption that the covariate is encoded via $z_c$ and that such a source of variability could be considered far out of distribution. Ultimately, it boils down to the quality of the data and how the new cell type relates to the seen ones. Arguably, such a cell type should be encoded with an amortised version of the current cell line embedding.
>
> ### General
>
> >*abbreviations (CPA, MoA, DMSO etc.) used are not defined properly*
>
> - We address inconsistencies concerning our use of abbreviations [L37, L81, L164-166, L186].
>
> >*difficult to know which label corresponds to which point*
>
> - We agree that Fig.3 is difficult to interpret. We aimed to demonstrate that users have access to the learned latent space by visualisation. The shown clustering is related to the drugs' effects as measured in the sci-Plex3 data. Hence, one cannot expect all drugs to cluster according to their MoA. Ideally, Fig.3 would be seen together with the provided uncertainty score. We plan to update Fig.3 but have not figured out how to improve it in the best possible way.

---

> > ### Comment · Reviewer_32go · 2022-08-03
> > **Thanks for a thorough reply.**
> >
> > Overall, my curiosity is very satisfied with the authors' response and I find the new version of the manuscript fixes most of the shortcomings that were pointed out by the reviewers.
> > Although some comments about the modelling choices can feel a bit underwhelming, I find that the current manuscript gives more insights and also reveals more clearly parts where chemCPA could maybe be further improved or built upon, which I think is very valuable for the research community.
> > To be honest, I am still a little bit confused about the architecture surgery and its details. And I am still as puzzled about the gradient penalty term. The authors removed the reference to WGAN-GP, which I agree was misleading, but there is now basically not a single insight or explanation beyond "it gives better results". I am aware that the CPA paper, from which it is taking its inspiration also provides not a single comment on this choice.
> > If you have more comments about this, I would be curious to read them. But I won't make it a deal breaker for the acceptance.
> >
> > I am waiting a bit longer to see if some discussion with other reviewers emerges but I will definitely update and improve my rating of the paper.
> > Good luck finding a way to improve Fig.3!

---

> > > ### Author Response · Authors · 2022-08-08
> > > **Further updates and clarifications**
> > >
> > > Thanks again for the quick feedback and positive evaluation.
> > >
> > > > *more comments about the gradient penalty*
> > >
> > > We checked again the GAN literature and now provide a more satisfying explanation. In chemCPA, we employ a penalty on the zero-centred gradient (0-GP), more specifically the squared L2-norm of the gradients wrt. the latent basal state $z_i$. [Mescheder, 2018](https://proceedings.mlr.press/v80/mescheder18a/mescheder18a.pdf) proves how this penalty, applied to GAN discriminators, enables local convergence under some assumptions and leads to more stable convergence. The penalty approximates Gaussian noise regularisation on $z_i$ (see [Roth et al, 2017](https://arxiv.org/pdf/1705.09367.pdf)). We have updated the manuscript accordingly and fixed an error in the equation [L147-148]. The penalty is now correctly stated as: $$L_{pen}^j = \frac{1}{k}\sum_k \big \lVert \partial_{z_i} A^j_\phi(z_i)_k \big \rVert_2^2  \: $$
> > >
> > > > *confused about the architecture surgery and its details*
> > >
> > > We tried to improve the explanation of the architecture surgery in the manuscript and added mathematical expressions to make it more precise.
> > >
> > > Note that scArches is a conditional VAE, and their architecture surgery refers to adding additional variables that are being conditioned on both by the encoder and decoder, details can be found [here](https://www.nature.com/articles/s41587-021-01001-7#Sec10). In contrast, chemCPA is a standard autoencoder. Our architecture surgery refers to adding two new non-linear layers, $h_\text{enc}: \mathbb{R}^{n_\text{finetune}}\rightarrow \mathbb{R}^{n_\text{pretrain}}$ and $h_\text{dec}:\mathbb{R}^{2n_\text{pretrain}}\rightarrow \mathbb{R}^{2n_\text{finetune}}$ to the autoencoder. Consequently, the encoder becomes $\hat E_{\theta} = E_\theta(h_\text{enc}(x))$ and the decoder becomes $\hat D_{\psi} = h_\text{dec}(D_\psi(z'))$. In the L1000 $\rightarrow$ SciPlex3 example: $n_\text{pretrain}=977, n_\text{finetune}=2000$. Therefore, we do not add individual neurons like scArches, but whole layers, and we also do not perform a transfer of dataset labels, but of gene sets.
> > >
> > > > *Figure 3*
> > >
> > > We have updated the figure to make it clear which labels correspond to which points. We additionally replaced the low dosage visualisation with the uncertainty score from section 5.3 to provide more context for the embeddings $z_d$.

---

> > > > ### Comment · Reviewer_32go · 2022-08-09
> > > > **it's more clear now**
> > > >
> > > > Thanks again to the authors for the further precisions.
> > > > It is not exactly clear (to me) to which extent the analysis of from Mescheder et al. and Roth et al. might apply to ChemCPA but deriving such an analysis would clearly fall out of the scope of the current paper.
> > > > With those further explanations the surgery procedure sounds very straightforward but I really had not understood before. So I hope those added explanations will make the manuscript more clear to the general audience.
> > > > Figure 3 is also much more readable now.
> > > > I'm updating my rating of the paper from an initial 5 to 8 and hope this paper will be accepted.

---

### Official Review · Reviewer_Qepn · 2022-07-11

**Rating:** 5
**Confidence:** 4
**Soundness:** 2 fair
**Presentation:** 3 good
**Contribution:** 3 good

**Summary:**

The authors describe a method for predicting the gene expression response of a cell line to drug treatment. They make use of an autoencoder framework, encoding the gene expression profile using a dense neural network layer. They use adversarial losses to discourage the model from encoding the cell line and drug treatment in the latent space. Finally, they derive features from the molecular structure of the drug using various approaches. Embeddings of the expression profile, cell line, and drug molecule are decoded with dense neural network layers and compared to the observed gene expression from the experiment by a loss function.

Aug 9 - I increased my ratings after reading the authors' revision and rebuttal.

**Questions:**

Line 90, Gaussian likelihood is inappropriate for RNA-seq.
Line 141, L1000 isn’t bulk RNA-seq, it’s a platform called NanoString.
Figure 3, in the caption, why are latent embeddings “identical” for molecules that target the same pathway?
Table 4, caption incorrectly describes table.

**Limitations:**

Yes, limitations have been addressed.

**Strengths And Weaknesses:**

Predicting cell responses to drugs is an important problem.

I don’t understand the evaluation. Why is the term “counterfactual” prediction used? It sounds like it refers to model encoding and decoding with DMSO as the molecule, but I don’t understand why this would be relevant. Are the authors evaluating performance on other molecules than DMSO? I’m confused by the R2 metric, which seems to be computed across genes for each sample, and the authors use of the term “disentanglement”. The authors describe Table 1 as if it evaluates disentanglement, but the metric is R2. What are the drug and cell line columns of Table 1 showing? In the case of DEGs, do the authors consider distinct DEGs for each molecule?

These ideas have been published previously. The authors don’t distinguish their method, and they don’t compare to previously published methods. E.g. see Lotfollahi, M. et al. Learning interpretable cellular responses to complex perturbations in high-throughput screens. bioRxiv 2021.04.14.439903 (2021) doi:10.1101/2021.04.14.439903. https://www.biorxiv.org/content/10.1101/2021.04.14.439903v2

---

> ### Author Response · Authors · 2022-08-02
> **Addressing concerns regarding the clarity of the performed evaluation and the missing comparison to existing methods**
>
> We thank the reviewer for their time and feedback. The reviewer raised two major concerns regarding the clarity of the performed evaluation and the missing comparison to already existing methods. We addressed both in our updated manuscript and give detailed answers to the raised questions in the section below.
>
> ### Counterfactual predictions
>
> #### chemCPA's design enables counterfactual predictions
>
> >*Why term “counterfactual”?*
>
> - The compositional nature of chemCPA, like CPA, allows asking questions akin to “What would be the perturbational response of cell $c$ if it had been perturbed with drug $d$ instead?”, and answering such questions at cellular resolution. We enforce this compositional latent structure through the adversarial loss. We refer to such predictions as counterfactual predictions and note that the term “counterfactual” may have different meanings in different sub-fields of machine learning [L185-188].
>
> >*...encoding and decoding with DMSO, but relevance unclear*
>
> - Cells that were treated with DMSO are the control cells and are considered unperturbed. This is common practice for perturbation experiments. To compute counterfactual predictions, we, therefore, first encode control cells and then add a perturbation vector $z_d$, corresponding to some drug $d$. This representation can belong to any other molecule $d$, potentially being absent from the training data $(d\notin\mathcal{D})$. The combined latent representation entails the drug’s perturbation and is the input of the decoder [L186-188]. Hence, we answer the question: “What would the cell's gene response look like, had it been treated with drug $d$ instead of DMSO?”.
>
> ### Evaluation
>
> #### The R2 score is the main evaluation metric, computed on all genes and the DEGs, chemCPA models are required to be disentangled
>
> >*Confused by the R2 metric*
>
> - Our performance measure is the R2 metric. This metric enables the comparison of chemCPA's counterfactual predictions with real experimental measurements [L189-197]. Following our explanation of counterfactual predictions, we use control cells (DMSO treated) for the evaluation. That is, we do not encode cells that were treated with another drug (although technically possible). For decoding, however, we consider many different drugs [L217-220]. Specifically, we choose nine different drugs to be fully ood (absent from the train and val set) for our experiments, cf. general reply. We compute the R2 score on the decoded output.
>
> >*...and use of the term “disentanglement”*
>
> - To identify perturbation effects, two conditions must be met. First, drug and cell line embeddings should represent what they are supposed to encode. Second, the basal state has to be uninformative for these two attributes. If these conditions hold, we consider the basal state $z_i$, the drug perturbation $z_d$, and the cell line embedding $z_c$ to be disentangled [L140-141]. To quantify this, we compute classification accuracies of non-linear MLPs that were trained separately from the adversarial classifiers and for each attribute individually [L198-200].
> - *What is Table 1 showing?*
>   - The drug and cell line columns in Table 1 (now Appendix) report such disentanglement scores (class. acc.) and the last two columns report R2 scores.
>
> >*Evaluating performance on other molecules than DMSO?*
>
> - We compare counterfactual predictions to real gene expression profiles using the R2 metric, across 9 hold-out drugs. This procedure makes the change in prediction identifiable for the perturbation encoding $z_d$ and, therefore, is relevant for the biological application. We updated the manuscript to improve clarity [Sec.5.1-5.3].
>
> >*Consider distinct DEGs for each molecule?*
>
> - The computation of DEGs was done with scanpy’s `rank_genes_groups` and is both dependent on the molecule and cell line. We always consider 50 DEGs [L194-197].
>
> ### Comparison to existing methods
>
> #### Our model, chemCPA, outperforms CPA and scGen on simpler tasks and allows to generalise to unseen drugs
>
> >*Don’t compare*
>
> - We now compared chemCPA to CPA and scGen. For this, we had to simplify the experiment to the generalisation of unobserved combinations of drugs and cell lines. ChemCPA outperforms both scGen and CPA, demonstrating that the perturbation network and pretraining lead to SOTA performance, cf. the general reply for details [Sec.5.1].
>
> >*Don’t distinguish*
>
> - While CPA requires the drug $d$ to be part of the training dataset, chemCPA can generalise to any drug via its perturbation network, even molecules that have not been previously observed anywhere $(d\notin\mathcal{D})$. To legitimise this generalisation, we evaluate chemCPA in the setting of transfer-learning [L48-51,L128-130,Sec. 5.1 - 5.3].
>
> ### General
>
> - We updated the manuscript to address our mistake on the L1000 data being RNA-seq.
> - We updated Fig.3. To clarify, $z_d$ is not identical across drugs, only for different dosage values.

---

> > ### Author Response · Authors · 2022-08-08
> > **Request for feedback**
> >
> > Dear reviewer,
> >
> > We would like to thank you once again for your time and feedback.
> > As we believe we made several important points worth discussing or acknowledging and were able to substantially improve our paper, we would like to kindly request further feedback as the deadline for the end of the author-reviewer discussion period approaches.
> >
> > Best regards,
> > the authors

---

> > > ### Comment · Area_Chair_J6Zu · 2022-08-09
> > > **Reviewer: Please give authors feedback on their answer**
> > >
> > > The authors have made quite some effort to answer your concerns.
> > >
> > > Your AC

---

> > > > ### Comment · Reviewer_Qepn · 2022-08-09
> > > > **Concerns mostly addressed**
> > > >
> > > > The model evaluations are much clearer now, and the manuscript has been improved.
> > > >
> > > > One remaining concern is your decision to test on only nine held out molecules. (I assume you're training on all 150+ ?) Given that an important novel aspect of your work is the molecule encoder, I'd recommend a much more thorough evaluation of the ability to generalize to unseen molecules. I can't imagine training these models is terribly compute intensive. I would've divided the molecules into folds and performed cross-validation to allow you access to 10x more test molecules. As is, the model's ability to identify molecule features that determine their influence on the transcriptome is not well established. Given the limitations of your current evaluation, I'll only bump my rating by one point.
> > > >
> > > > Minor comments:
> > > > Line 171, should be “Ensembl gene annotations”
> > > > Table 1, specify dataset.

---

### Official Review · Reviewer_mn3q · 2022-07-12

**Rating:** 7
**Confidence:** 3
**Soundness:** 3 good
**Presentation:** 3 good
**Contribution:** 3 good

**Summary:**

In this paper, the authors propose a new model called chemCPA to study the perturbational effects of unseen drugs by incorporating knowledge about the drug's molecular structure.   They introduce new encoder-decoder architecture and benchmark against other molecule encoding networks. Also, the authors combine this model with a transfer learning scheme that improves generalization by utilizing existing HTS bulk RNA-seq data.

**Questions:**

1) I am not entirely clear on how subsetting of 977 genes is done via ensemble gene annotations?
2) have the authors looked into other datasets mentioned in the Compositional perturbation autoencoder paper?

**Limitations:**

Limitations are scattered across multiple sections. It may be much more helpful to assemble them in the discussion section for improved readability

**Strengths And Weaknesses:**

The authors propose a model for predicting cellular gene expression responses for unseen drug perturbations by encoding the drugs’ molecular structures. The approach is well motivated and is described clearly. Authors have demonstrated that the pretraining on the abundance of bulk HTS datasets improves generalization. The chemCPA has several advantages like a latent space disentanglement and the induced interpretability of the perturbations in the context of cell state and covariates, incorporating molecular structure over regular encoders.

One concern is that the performance of chemCPA models on DEGs is worse compared to all genes. Also, it may be beneficial to benchmark chemCPA against CPA or scGen in less complicated scenarios with a few perturbations.

On line 60, computational perturbation auto-encoder (CPA) should be changed to Compositional perturbation autoencoder

---

> ### Author Response · Authors · 2022-08-02
> **Including a new comparison experiment and improving structure of manuscript for clarity**
>
> We thank the reviewer for the positive evaluation and feedback on our work. We will now address the remaining concerns one by one and update our manuscript accordingly. Most importantly, we included a new experiment comparing chemCPA to CPA and scGen (see general reply to all reviewers).
>
> ### Performance evaluation
>
> #### The performance on DEGs is expected to be worse compared to the whole gene set
>
> >*One concern is that the performance of chemCPA models on DEGs is worse compared to all genes.*
>
> - It is generally expected that the predictive performance on the subset of differentially expressed genes (DEGs) is worse compared to the whole gene set. Most genes do not change much when a cell is treated with a drug. This explains why a model that simply discards any information about drug perturbations still achieves good R2 scores when measured across the whole gene set. Therefore, to discern model performance, we focus our evaluations also on the DEGs [L190-197, L253]. We report the perturbation-discarding model as our baseline and understand the improvement of chemCPA over the baseline as a result of the additional drug encoding.
> - *Remark*: For unseen drugs, the DEGs are unknown as they depend on the drug and cell type. As a consequence, it is essential to report both the scores on all genes as well as the DEGs. The combination of the two is important to gauge the accuracy of the model’s predictions [L194-197].
>
> ### Transfer learning
>
> #### Pretraining requires a shared set of genes between source (L1000) and target (sciPlex3)
>
> >*not entirely clear on how subsetting of 977 genes is done via ensemble gene annotation*
>
> - For the transfer-learning part of chemCPA, the models must share at least some genes. As correctly suggested, we identified the set of shared genes between the L1000 and the sciPlex3 experiment via gene ids [L169-170]. As the limitation to a specific set of genes is very restrictive, especially in the single-cell setting, we consider two settings (5.2 and 5.3) in our manuscript:
>   - The first considers all 977 genes that are present in L1000 and sciPlex3,
>   - The second considers an additional 1023 highly-variable genes that were only observed in sciPlex3, yielding 977+1023 genes in total.
>
> ### General
>
> >*Have the authors looked into other datasets mentioned in the Compositional perturbation autoencoder paper?*
>
> - The perturbation datasets that CPA was evaluated on were either small molecule perturbations (included in chemCPA) or CRISPR gene perturbations. The gene perturbation datasets can not be represented with the new perturbation network $P_\varphi$ and do not match the focus of this work. Therfore, we did not include them. There are other approaches specifically designed for modeling CRISPR pertrubations like [GEARS](https://www.biorxiv.org/content/10.1101/2022.07.12.499735v2).
>
> >*Limitations are scattered across multiple sections. It may be much more helpful to assemble them in the discussion section for improved readability*
>
> - We consolidated the information about the limitations at the beginning of Sec.5.3 [L283-291].

---

> > ### Comment · Reviewer_mn3q · 2022-08-07
> > **Thanks**
> >
> > The authors have addressed all my concerns/questions. I have updated my rating

---

### Author Response · Authors · 2022-08-02
**Including a new experiment that compares chemCPA to existing methods**

## General comment to all reviewers

We thank all reviewers for their valuable feedback on multiple crucial parts of our manuscript. Most importantly, a ***comparison experiment*** of chemCPA to already existing methods in a less ambitious setting was missing. As this experiment was requested by all reviewers, we present our results in this general answer. We also comment on our choice of reconstruction loss as this was requested by multiple reviewers.

In addition, we give detailed answers to each reviewer and point towards ***changes in our manuscript which are highlighted in blue in the pdf***:

- R2 raised several points regarding our model design to enable ***counterfactual predictions*** and the connection to the performed ***evaluation*** of chemCPA
- R3  raised several points on the ***modelling assumptions*** of chemCPA
- R1 & R3 were both interested in clarifications on the ***transfer learning*** part of chemCPA

### Additional comparison experiment

For this additional comparison of chemCPA to existing methods, we had to choose a less ambitious setting, as only chemCPA is capable of predicting gene expressions for previously unseen drugs. As R3 suggested, we compared CPA, scGen, and chemCPA on their ability to generalize to unobserved combinations of drugs and cell lines [Section 5.1].

As scGen cannot distinguish between different dosage values, we performed two separate experiments for the second and highest dose values, respectively. Moreover, as both CPA and scGen require each individual component (drug, cell line) to be part of the training data, we created a split for each of the three cell lines according to:

| Drug-covariate combination | Split 1 | Split 2 | Split 3 |
| :------------------------: | :-----: | :-----: | :-----: |
| $\{\text{Drugs } i\}$, A549  | Training, Validation | Training, Validation | Test |
| $\{\text{Drugs } i\}$, K562  | Training, Validation | Test | Training, Validation |
| $\{\text{Drugs } i\}$, MCF7  | Test | Training, Validation | Training, Validation |

The set of drugs $i$ is the same as presented for the experiments in sections 5.2 and 5.3 [L217-218]:

- Dacinostat, Givinostat, Belinostat, Hesperadin, Quisinostat, Alvespimycin, Tanespimycin, TAK-901, and Flavopiridol.

Following our other experiments on unseen drugs, we report mean and median R2 values (averaged over the 3 splits and all drugs $i$) for the sets of all and differentially expressed genes (DEGs). The baseline is again a model that ignores drugs completely.

#### Experiment result

**Dose:** $1.0\,\mu$Mol

| Model | Median R2 all genes | Median R2 DEGs | Mean R2 all genes | Mean R2 DEGs |
|:------|:-------------------:|:--------------:|:-----------------:|:------------:|
| Baseline           |   82.5 |    62.2   |   69.3   |   51    |
| scGen              |   76.7 |    68.1   |   72.7   |   59.1  |
| CPA                | **85.6** | 66.8   |   71.7   |   54.1  |
| chemCPA            | **85.7** | 66.1   |   73.8   |   60.5  |
| chemCPA pretrained |   85.3 | **75.5** | **76.9** | **68.3**|

**Dose**: $10\,\mu$Mol

| Model | Median R2 all genes | Median R2 DEGs | Mean R2 all genes | Mean R2 DEGs |
|:------|:-------------------:|:--------------:|:-----------------:|:------------:|
| Baseline           |   48.5   |   12.2   |   50.3   |    28.7   |
| scGen              |   66.2   |   48.9   |   62.2   |    47.2   |
| CPA                |   52.5   |   25.7   |   53.9   |    34.2   |
| chemCPA            |   76.7   |   64.4   |   71.2   |    58.4   |
| chemCPA pretrained | **82.4** | **78.6** | **76.5** |  **67.8** |

To make a fair comparison, we optimised all CPA and chemCPA models identically and swept over the same set of hyperparameters (random, 10 samples). scGen was optimised with default parameters provided by scVI-tools and an adjusted KL annealing scheme to match the set number of epochs. Note that both CPA and chemCPA can take control cells $x_i$ from all cell lines as input $(\{\text{A549, K562, MCF7}\}\rightarrow\{\text{A549 K562, MCF7}\})$, while the cell line input for scGen has to match the test set $(\text{A549}\rightarrow\text{A549},\text{K562}\rightarrow\text{K562}, \text{MCF7}\rightarrow\text{MCF7})$.

### Gaussian Likelihood and count data

We agree that for count data like scRNA-seq, a count distribution like Poisson or Negative Binomial is appropriate. The used Gaussian likelihood becomes applicable when the data is normalised and $\log 1$ transformed [L167-168]. Our code also supports the NB loss; however, we observed better convergence for chemCPA with the Gaussian loss, which is why we stuck to this for the manuscript. We clarified this reasoning and updated the manuscript accordingly [L92-95]. We are happy to include empirical comparisons to using the NB loss in the supplement upon request.

---

### Meta-Review · Area_Chair_J6Zu · 2022-08-27

**Recommendation:** Accept
**Confidence:** Certain

**Metareview:**

All four reviewers liked aspects of the paper with one still on the fence. All reviewers appreciated the authors' feedback to the their comments both in the discussion and the extensive updates of the paper.

Accept is recommended.

**Award:**

No

---

### Decision · Program_Chairs · 2022-09-14

Accept